# TOWARDS CONVERGENCE TO NASH EQUILIBRIA IN TWO-TEAM ZERO-SUM GAMES

**Fivos Kalogiannis**
UC Irvine

**Ioannis Panageas**
UC Irvine

**Emmanouil V. Vlatakis-Gkaragkounis**
Columbia University

## ABSTRACT

Contemporary applications of machine learning in two-team e-sports and the superior expressivity of multi-agent generative adversarial networks raise important and overlooked theoretical questions regarding optimization in two-team games. Formally, two-team zero-sum games are defined as multi-player games where players are split into two competing sets of agents, each experiencing a utility identical to that of their teammates and opposite to that of the opposing team. We focus on the solution concept of Nash equilibria (NE). We first show that computing NE for this class of games is *hard* for the complexity class CLS. To further examine the capabilities of online learning algorithms in games with full-information feedback, we propose a benchmark of a simple —yet nontrivial— family of such games. These games do not enjoy the properties used to prove convergence for relevant algorithms. In particular, we use a dynamical systems perspective to demonstrate that gradient descent-ascent, its optimistic variant, optimistic multiplicative weights update, and extra gradient fail to converge (even locally) to a Nash equilibrium. On a brighter note, we propose a first-order method that leverages control theory techniques and under some conditions enjoys last-iterate local convergence to a Nash equilibrium. We also believe our proposed method is of independent interest for general min-max optimization.

## 1 INTRODUCTION

Online learning shares an enduring relationship with game theory that has a very early onset dating back to the analysis of *fictitious play* by (Robinson, 1951) and Blackwell's *approachability theorem* (Blackwell, 1956). A key question within this context is whether self-interested agents can arrive at a game-theoretic *equilibrium* in an *independent* and *decentralized* manner with only *limited feedback* from their environment. Learning dynamics that converge to different notions of equilibria are known to exist for two-player zero-sum games (Robinson, 1951; Arora et al., 2012; Daskalakis et al., 2011), potential games (Monderer & Shapley, 1996), near-potential games (Anagnostides et al., 2022b), socially concave games (Golowich et al., 2020), and extensive form games (Anagnostides et al., 2022a). We try to push the boundary further and explore whether equilibria —in particular, Nash equilibria— can be reached by agents that follow decentralized learning algorithms in two-team zero-sum games.

*Team competition* has played a central role in the development of game theory (Marschak, 1955; von Stengel & Koller, 1997; Bacharach, 1999; Gold, 2005), economics (Marschak, 1955; Gottinger, 1974), and evolutionary biology (Nagylaki, 1993; Nowak et al., 2004). Recently, competition among teams has attracted the interest of the machine learning community due to the advances that multi-agent systems have accomplished: *e.g.*, multi-GAN's (Hoang et al., 2017; Hardy et al., 2019) for generative tasks, adversarial regression with multiple learners (Tong et al., 2018), or AI agents competing in e-sports (*e.g.*, CTF (Jaderberg et al., 2019) or Starcraft (Vinyals et al., 2019)) as well as card games (Moravčík et al., 2017; Brown & Sandholm, 2018; Bowling et al., 2015).

**Our class of games.** We turn our attention to *two-team zero-sum games* a quite general class of min-max optimization problems that include bilinear games and a wide range of nonconvex-nonconcave games as well. In this class of games, players fall into two teams of size $n, m$ and submit their own randomized strategy vectors independently. We note that the games that we focus on are

not restricted to team games in the narrow sense of the term "team" as we use it in sports, games, and so on; the players play independently and do not follow a central coordinating authority. Rather, for the purpose of this paper, *teams* are constituted by agents that merely enjoy the same utility function. This might already hint that the solution concept that we engage with is the *Nash equilibrium* (NE). Another class of games that is captured by this framework is the class of *adversarial potential games*. In these games, the condition that all players of the same team experience the same utility is weakened as long as there exists a *potential function* that can track differences in the utility of each player when they unilaterally deviate from a given strategy profile (see Appendix A.2 for a formal definition). A similar setting has been studied in the context of nonatomic games (Babaioff et al., 2009).

**Positive duality gap.** In two-player zero-sum games, *i.e.*, $n = m = 1$, min-max (respectively max-min) strategies are guaranteed to form a Nash equilibrium due to Von Neumann's minmax theorem (Von Neumann, 1928); ultimately endowing the game with a unique value. The challenges arise for the case of $n, m > 1$; Schulman & Vazirani (2019b) prove that, in general, two-team games do not have a unique value. They do so by presenting a family of team games with a positive duality gap, together with bounds concerning this gap. These bounds quantify the effect of exchanging the order of commitment to their strategy either between the teams as a whole or the individual players.

**Solution concept.** In this work, we examine the solution concept of Nash equilibrium (NE). Under a Nash equilibrium, no player can improve their utility by unilaterally deviating. The main downside of a NE for team games is the fact that such an equilibrium can be arbitrarily suboptimal for the team (Basilico et al., 2017a).

This is one of the reasons that the solution concept of team-maxmin equilibrium with a coordination device (TMECor) has dominated contemporary literature of team games, especially in regard to applications (Farina et al., 2018; Zhang et al., 2020; Cacciamani et al., 2021). Under a TMECor, players are allowed to communicate before the game and decide upon combinations of strategies to be played during the game using an external source of randomness.

The undeniable advantage of a TMECor is that the expected utility of the team under it is greater than the expected utility under a NE (Basilico et al., 2017a). Nevertheless, this favorable property of TMECor can by no means render the study of NE irrelevant. In fact, the study of NE is always of independent interest within the literature of algorithmic game theory —especially questions corresponding to computational complexity. Moreover, there exist settings in which *ex ante* coordination cannot be expected to be possible or even sensible; for example in (i) environments where the external sources of randomness are unreliable or nonexistent or visible to the adversarial team, (ii) games in which players cannot know in advance who they share a common utility with, (iii) *adversarial potential games*. These games can model naturally occurring settings such as (a) security games with multiple uncoordinated defenders versus multiple similarly uncoordinated attackers, (b) the load balancing "game" between telecommunication service providers trying to minimize the maximum delay of service experienced by their customers versus the service users that try to individually utilize the maximum amount of broadband possible, and (c) the *weak selection* model of evolutionary biology where a species as a whole is a team, the genes of its population are the players and the alleles of each gene are in turn the actions of a player; the allele frequencies are independent across genes (Nagylaki, 1993; Nowak et al., 2004; Mehta et al., 2015).

Concluding, we could not possibly argue for a single correct solution concept for two-team games; there is no silver bullet. In contrast, one has to assess which is the most fitting based on the constraints of a given setting. A Nash equilibrium is a cornerstone concept of game theory and examining its properties in different games is always important.

**The optimization point of view.** We focus on the solution concept of NE and we first note that computing local-NE in general nonconvex-nonconcave games is PPAD-complete (Daskalakis et al., 2009; 2021). Thus, all well-celebrated online learning, first-order methods like gradient descent-ascent (Lin et al., 2020; Daskalakis & Panageas, 2019), its optimistic (Popov, 1980; Chiang et al., 2012; Sridharan & Tewari, 2010), optimistic multiplicative weights update (Sridharan, 2012), and the extra gradient method (Korpelevich, 1976) would require an exponential number of steps in the parameters of the problem in order to compute an approximate NE under the oracle optimization model of (Nemirovskij & Yudin, 1983). Additionally, in the continuous time regime, similar classes

of games exhibit behaviors antithetical to convergence like cycling, recurrence, or chaos (Vlatakis-Gkaragkounis et al., 2019). Second, even if a regret notion within the context of team-competition could be defined, no-regret dynamics are guaranteed to converge only to the set of coarse correlated equilibria (CCE) (Fudenberg, 1991; Hannan, 2016). CCE is a weaker equilibrium notion whose solutions could potentially be exclusively supported on strictly dominated strategies, even for simple symmetric two-player games (See also (Viossat & Zapechelnyuk, 2013)).

Surely, the aforementioned intractability remarks for the general case of nonconvex-nonconcave min-max problems provide a significant insight. But, they cannot *per se* address the issue of computing Nash equilibria when the game is equipped with a particular structure, *i.e.*, that of two-team zero-sum games. In fact, our paper addresses the following questions:

*Can we get provable convergence guarantees to NE of decentralized first-order methods in two-team zero-sum games?*

**Our results.** First, with regards to computational complexity, we prove that computing an approximate (and possibly mixed) NE in two-team zero-sum games is CLS-hard (Theorem 3.1); *i.e.*, it is computationally harder than finding pure NE in a congestion game or computing an approximate fixed point of gradient descent.

Second, regarding online learning for equilibrium computation, we prove that a number of established, decentralized, first-order methods are not fit for the purpose and fail to converge even asymptotically. Specifically, we present a simple —yet nontrivial— family of two-team zero-sum games (with each team consisting of two players) where projected gradient descent-ascent (GDA), optimistic gradient descent-ascent (OGDA), optimistic multiplicative weights update (OMWU), and the extra gradient method (EG) fail to locally converge to a mixed NE (Theorem 3.3). More broadly, in the case of GDA in nondegenerate team games with unique mixed NE, one could acquire an even stronger result for any high-dimensional configuration of actions and players (Theorem 3.2). To the best of our knowledge, the described family of games is the first-of-its-kind in which all these methods provably fail to converge at the same time.

Third, we propose a novel first-order method inspired by adaptive control (Bazanella et al., 1997; Hassouneh et al., 2004). In particular, we use a technique that manages to stabilize unstable fixed points of a dynamical system without prior knowledge of their position and without introducing new ones. It is important to note that this method is a modification of GDA that uses a stabilizing feedback which maintains the decentralized nature of GDA.

Finally, in Section 4 we provide a series of experiments in simple two-team zero-sum GAN's. We also show that multi-GAN architectures achieve better performance than single-agent ones, relative to the network capacity when they are trained on synthetic or real-world datasets like CIFAR10.

**Existing algorithms for NE in multiplayer games.** The focus of the present paper is examining algorithms for the setting of *repeated games* (Cesa-Bianchi & Lugosi, 1999, Chapter 7). If we do not restrict ourselves to this setting, there are numerous centralized algorithms (Lipton et al., 2003; Berg & Sandholm, 2017) and heuristics (Gemp et al., 2021) that solve the problem of computing Nash equilibria in general multi-player games.

## 2 PRELIMINARIES

**Our setting.** A *two-team game* in normal form is defined as a tuple $\Gamma(\mathcal{N}, \mathcal{M}, \mathcal{A}, \mathcal{B}, \{U_A, U_B\})$. The tuple is defined by

(i) a finite set of $n = |\mathcal{N}|$ *players* belonging to team $A$, as well as a finite set of $m = |\mathcal{M}|$ *players* belonging to team $B$;

(ii) a finite set of *actions* (or *pure strategies*) $\mathcal{A}_i = \{\alpha_1, \ldots, \alpha_{n_i}\}$ per player $i \in \mathcal{N}$; where $\mathcal{A} := \prod_i \mathcal{A}_i$ denotes the ensemble of all possible action profiles of team $A$, and respectively, a finite set of *actions* (or *pure strategies*) $\mathcal{B}_i = \{\beta_1, \ldots, \beta_{n_i}\}$ per player $i \in \mathcal{M}$, where $\mathcal{B} := \prod_i \mathcal{B}_i$.

(iii) a utility function for team $A$, $U_A : \mathcal{A} \times \mathcal{B} \to \mathbb{R}$ (resp. $U_B : \mathcal{A} \times \mathcal{B} \to \mathbb{R}$ for team $B$)

We also use $\boldsymbol{\alpha} = (\alpha_1, \ldots, \alpha_n)$ to denote the strategy profile of team $A$ players and $\boldsymbol{\beta} = (\beta_1, \ldots, \beta_m)$ the strategy profile of team $B$ players.

Finally, each team's *payoff* function is denoted by $U_A, U_B : \mathcal{A} \times \mathcal{B} \to \mathbb{R}$, where the *individual utility* of a player is identical to her teammates, i.e., $U_i = U_A$ & $U_j = U_B$ $\forall i \in \mathcal{N}$ and $j \in \mathcal{M}$. In this general context, players could also submit *mixed strategies*, i.e, probability distributions over actions. Correspondingly, we define the product distributions $\boldsymbol{x} = (\boldsymbol{x}_1, \ldots, \boldsymbol{x}_n)$, $\boldsymbol{y} = (\boldsymbol{y}_1, \ldots, \boldsymbol{y}_m)$ as team $A$ and $B$'s strategies respectively, in which $\boldsymbol{x}_i \in \Delta(\mathcal{A}_i)$ and $\boldsymbol{y}_j \in \Delta(\mathcal{B}_j)$. Conclusively, we will write $\mathcal{X} := \prod_{i \in \mathcal{N}} \mathcal{X}_i = \prod_{i \in \mathcal{N}} \Delta(\mathcal{A}_i), \mathcal{Y} := \prod_{i \in \mathcal{M}} \mathcal{Y}_i = \prod_{i \in \mathcal{M}} \Delta(\mathcal{B}_i)$ the space of mixed strategy profiles of teams $A, B$. A two-team game is called *two-team zero-sum* if $U_B = -U_A = U$ which is the main focus of this paper. Moreover, we assume that the game is *succinctly representable* and satisfies the *polynomial expectation property* (Daskalakis et al., 2006). This means that given a mixed strategy profile, the utility of each player can be computed in polynomial time in the number of agents, the sum of the number of strategies of each player, and the bit number required to represent the mixed strategy profile.

A *Nash equilibrium* (NE) is a strategy profile $(\boldsymbol{x}^*, \boldsymbol{y}^*) \in \mathcal{X} \times \mathcal{Y}$ such that

$$\begin{cases} U(\boldsymbol{x}^*, \boldsymbol{y}^*) \leq U(\boldsymbol{x}_i, \boldsymbol{x}^*_{-i}, \boldsymbol{y}^*), \ \forall \boldsymbol{x}_i \in \mathcal{X}_i \quad _1 \\ U(\boldsymbol{x}^*, \boldsymbol{y}^*) \geq U(\boldsymbol{x}^*, \boldsymbol{y}_j, \boldsymbol{y}^*_{-j}), \ \forall \boldsymbol{y}_j \in \mathcal{Y}_j \end{cases} \tag{NE}$$

**A first approach to computing NE in Two-Team Zero-Sum games.** Due to the multilinearity of the utility and the existence of a duality gap, the linear programming method used in two-player zero-sum games cannot be used to compute a Nash equilibrium. For the goal of computing Nash equilibrium in two-team zero-sum games, we have experimented with a selection of online learning, first-order methods that have been utilized with varying success in the setting of the two-person zero-sum case. Namely, we analyze the following methods: (i) gradient descent-ascent (GDA) (ii) optimistic gradient descent-ascent (OGDA) (iii) extra gradient method (EG) (iv) optimistic multiplicative weights update method (OMWU). For their precise definitions, we refer to Appendix B.

The below folklore fact will play a key role hereafter.

**Fact 2.1.** Any fixed point of the aforementioned discrete-time dynamics (apart from OMWU) on the utility function necessarily corresponds to a Nash Equilibrium of the game.

Hence, an important test for the asymptotic behavior of GDA, OGDA, EG, and OMWU methods is to examine whether these methods stabilize around their fixed points which effectively constitute the Nash equilibria of the game. In Section 3.2, we show that in the absence of pure Nash equilibria, all the above methods fail to stabilize on their fixed points even for a simple class of two-team game with $(n = 2, m = 2)$. Consequently, they fail to converge to the mixed Nash equilibrium of the game.

The presence of these results demonstrates the need for a different approach that lies outside the scope of traditional optimization techniques. Inspired by the applications of washout filters to stabilize unknown fixed points and the adaptive control generalizations of the former, we design a new variant of GDA "vaned" with a feedback loop dictated by a pair of two matrices. In contrast to the aforementioned conventional methods, our proposed technique surprisingly accomplishes asymptotic last-iterate convergence to its fixed point, *i.e.*, the mixed Nash equilibria of the team game.

$(\mathbf{K}, \mathbf{P})$**-vaned GDA Method.** After concatenating the vectors of the minimizing and the maximizing agents — $\boldsymbol{z}^{(k)} = (\boldsymbol{x}^{(k)}, \boldsymbol{y}^{(k)})$ — our method for appropriate matrices $\mathbf{K}, \mathbf{P}$ reads:

$$\begin{cases} \boldsymbol{z}^{(k+1)} = \Pi_{\mathcal{Z}} \Big\{ \boldsymbol{z}^{(k)} + \eta \big( \begin{smallmatrix} -\nabla_{\boldsymbol{x}} f(\boldsymbol{z}^{(k)}) \\ \nabla_{\boldsymbol{y}} f(\boldsymbol{z}^{(k)}) \end{smallmatrix} \big) + \eta \mathbf{K}(\boldsymbol{z}^{(k)} - \boldsymbol{\theta}^{(k)}) \Big\} \\ \boldsymbol{\theta}^{(k+1)} = \Pi_{\mathcal{Z}} \Big\{ \boldsymbol{\theta}^{(k)} + \eta \mathbf{P}(\boldsymbol{z}^{(k)} - \boldsymbol{\theta}^{(k)}) \Big\} \end{cases} \tag{KPV-GDA}$$

Intuitively, the additional variable $\boldsymbol{\theta}^{(k)}$ holds an estimate of the fixed point, and through the feedback law $\eta \mathbf{K}(\boldsymbol{z}^{(k)} - \boldsymbol{\theta}^{(k)})$ the vector $\boldsymbol{z}$ stabilizes around that estimate which slowly moves towards the real fixed point of the GDA dynamic.

### 2.1 TWO ILLUSTRATIVE EXAMPLES

Our first example plays a dual role: first, it demonstrates how two-team min-max competition can capture the formulation of multi-agent GAN architectures; second, it hints at the discrepancy be-

---

[1] We are using here the shorthand $\boldsymbol{x}_{-i}$ (or $\boldsymbol{y}_{-i}$) to highlight the strategy of all agents $\mathcal{N}$ (or $\mathcal{M}$) but $i$.

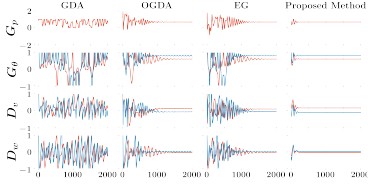

Figure 1: Parameter training of the configuration under different algorithms

tween the results of optimization methods, since —as we will see— GDA will not converge to the Nash equilibrium/ground-truth distribution. Generally, the solution that is sought after is the $\min\max$ solution of the objective function (Goodfellow et al., 2014) which are NP-hard to compute in the general case (Borgs et al., 2008); nevertheless, applications of GANs have proven that first-order stationary points of the objective function suffice to produce samples of very good quality.

### 2.1.1 LEARNING A MIXTURE OF GAUSSIANS WITH MULTI-AGENT GAN'S

Consider the case of $\mathcal{O}$, a mixture of gaussian distribution with two components, $C_1 \sim \mathcal{N}(\boldsymbol{\mu}, \mathbf{I})$ and $C_2 \sim \mathcal{N}(-\boldsymbol{\mu}, \mathbf{I})$ and mixture weights $\pi_1, \pi_2$ to be positive such that $\pi_1 + \pi_2 = 1$ and $\pi_1, \pi_2 \neq \frac{1}{2}$.

To learn the distribution above, we utilize an instance of a *Team*-WGAN in which there exists a generating team of agents $G_p : \mathbb{R} \to \mathbb{R}, G_{\boldsymbol{\theta}} : \mathbb{R}^n \to \mathbb{R}^n$, and a discriminating team of agents $D_{\boldsymbol{v}} : \mathbb{R}^n \to \mathbb{R}, D_{\boldsymbol{w}} : \mathbb{R}^n \to \mathbb{R}$, all described by the following equations:

$$\begin{aligned} \text{Generators:} \quad & G_p(\zeta) = p + \zeta \ , \ G_\theta(\boldsymbol{\xi}) = \boldsymbol{\xi} + \boldsymbol{\theta} \\ \text{Discriminators:} \quad & D_{\boldsymbol{v}}(\boldsymbol{y}) = \langle \boldsymbol{v}, \boldsymbol{y} \rangle \ , \ D_{\boldsymbol{w}}(\boldsymbol{y}) = \sum_i w_i y_i^2 \end{aligned} \tag{1}$$

The generating agent $G_\theta$ maps random noise $\boldsymbol{\xi} \sim \mathcal{N}(0, \mathbf{I})$ to samples while generating agent $G_p(\zeta)$, utilizing an independent source of randomness $\zeta \sim \mathcal{N}(0, 1)$, probabilistically controls the sign of the output of the generator $G_\theta$. The probability of ultimately generating a sample $\boldsymbol{y} = \boldsymbol{\xi} + \boldsymbol{\theta}$ is in expectation equal to $p$, while the probability of the sample being $\boldsymbol{y} = -\boldsymbol{z} - \boldsymbol{\theta}$ is equal to $1 - p$.

On the other end, there stands the discriminating team of $D_{\boldsymbol{v}}, D_{\boldsymbol{w}}$. Discriminators, $D_v(\boldsymbol{y}), D_w(\boldsymbol{y})$ map any given sample $\boldsymbol{y}$ to a scalar value accounting for the "realness" or "fakeness" of it – negative meaning fake, positive meaning real. The discriminators are disparate in the way they measure the realness of samples as seen in their definitions.

We follow the formalism of the Wasserstein GAN to form the optimization objective:

$$\max_{\boldsymbol{v},\boldsymbol{w}} \min_{\boldsymbol{\theta},p} \left\{ \mathbb{E}_{\boldsymbol{y}\sim\mathcal{O}}\Big[D_{\boldsymbol{v}}(\boldsymbol{y}) + D_{\boldsymbol{w}}(\boldsymbol{y})\Big] - \mathbb{E}_{\substack{\boldsymbol{\xi}\sim\mathcal{N}(0,\mathbf{I}),\\ \zeta\sim\mathcal{N}(0,1)}} \left[ \begin{array}{c} G_p(\zeta)\cdot\Big(D_{\boldsymbol{v}}\big(G_{\boldsymbol{\theta}}(\boldsymbol{y})\big)+D_{\boldsymbol{w}}\big(G_{\boldsymbol{\theta}}(\boldsymbol{y})\big)\Big) \\ + \\ \big(1-G_p(\zeta)\big)\cdot\Big(D_{\boldsymbol{v}}\big(-G_{\boldsymbol{\theta}}(\boldsymbol{y})\big)+D_{\boldsymbol{w}}\big(-G_{\boldsymbol{\theta}}(\boldsymbol{y})\big)\Big) \end{array} \right] \right\} \tag{2}$$

Equation (2) yields the simpler form:

$$\max_{\boldsymbol{v},\boldsymbol{w}} \min_{\boldsymbol{\theta},p} (\pi_1 - \pi_2)\boldsymbol{v}^T\boldsymbol{\mu} - 2p\boldsymbol{v}^T\boldsymbol{\theta} + \boldsymbol{v}^T\boldsymbol{\theta} + \sum_i^n w_i(\mu_i^2 - \theta_i^2) \tag{3}$$

It is easy to check that Nash equilibria of (2) must satisfy:

$$\left\{ \begin{array}{llll} \boldsymbol{\theta} & = & \boldsymbol{\mu}, & p = 1 - \pi_2 = \pi_1 \\ \boldsymbol{\theta} & = & -\boldsymbol{\mu}, & p = 1 - \pi_1 = \pi_2. \end{array} \right\}$$

Figure 1 demonstrates both GDA's failure and OGDA, EG, and our proposed method, KPV-GDA succeeding to converge to the above Nash equilibria and simultaneously discovering the mixture of the ground-truth.

### 2.1.2 MULTIPLAYER MATCHING PENNIES

Interestingly enough, there are non-trivial instances of two-team competition settings in which even OGDA and EG fail to converge. Such is the case for a team version of the well-known game of

matching pennies. The game can be shortly described as such: "*coordinate with your teammates to play a game of matching pennies against the opposing team, coordinate not and pay a penalty*". (We note that this game is a special case of the game presented in Section 3.3.) As we can see in Figures 2a and 2b, this multiplayer generalized matching pennies game constitutes an excellent benchmark on which all traditional gradient flow discretizations fail under the perfect competition setting. Interestingly, we are not aware of a similar example in min-max literature and it has been our starting point for seeking new optimization techniques inspired by Control theory. Indeed, the KPV-GDA variation with $(\mathbf{K}, \mathbf{P}) = (-1.1 \cdot \mathbf{I}, 0.3 \cdot \mathbf{I})$ achieves to converge to the unique mixed Nash Equilibrium of the game. In the following sections, we provide theorems that explain formally the behavior of the examined dynamics.

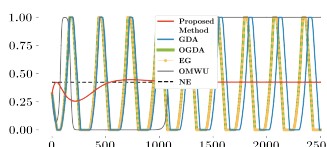

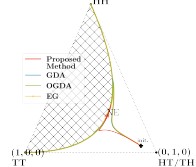

(a) Generalized matching pennies under different algorithms. For the precise definition of the game, we refer to appendix C.4

(b) Projected Trajectory of Team A under different algorithms. The sketched surface is not part of the feasible team strategy profiles (product of distributions).

## 3 MAIN RESULTS

In this section, we will prove that computing a Nash equilibrium in two-team zero-sum games is computationally hard and thus getting a polynomial-time algorithm that computes a Nash equilibrium is unlikely. Next, we will demonstrate the shortcomings of an array of commonly used online learning, first-order methods, and then we will provide a novel, decentralized, first-order method that locally converges to NE under some conditions.

### 3.1 ON THE HARDNESS OF COMPUTING NE IN TWO-TEAM ZERO-SUM GAMES

As promised, our first statement characterizes the hardness of NE computation in two-team zero-sum games:

**Theorem 3.1** (CLS-hard). *Computing a Nash equilibrium in a succinctly represented two-team zero-sum game is* CLS-*hard.*

The main idea of the proof of Theorem 3.1 relies on a reduction of approximating Nash equilibria in congestion games, which has been shown to be complete for the CLS complexity class. The class CLS contains the problem of continuous optimization. We defer the proof of the above theorem to the paper's supplement.

### 3.2 FAILURE OF COMMON ONLINE, FIRST-ORDER METHODS

The negative computational complexity result we proved for two-team zero-sum games (Theorem 3.1) does not preclude the prospect of attaining algorithms (learning first-order methods) that converge to Nash equilibria. Unfortunately, we prove that these methods cannot guarantee convergence to Nash equilibria in two-team zero-sum games in general.

In this subsection, we are going to construct a family of two-team zero-sum games with the property that the dynamics of GDA, OGDA, OMWU, and EG are unstable on Nash equilibria. This result is indicative of the challenges that lie in the min-max optimization of two-team zero-sum games and the reason that provable, nonasymptotic convergence guarantees of online learning have not yet been established.

Before defining our benchmark game, we prove an important theorem which states that GDA does not converge to mixed Nash equilibria. This fact is a stepping stone in constructing the family of team-zero sum games later. We present the proof of all of the below statements in detail in the paper's appendix (Appendix B).

**Weakly-stable Nash equilibrium.**  (Kleinberg et al., 2009; Mehta et al., 2015) Consider the set of Nash equilibria with the property that if any single randomizing agent of one team is forced to play any strategy in their current support with probability one, all other agents of the same team must remain indifferent between the strategies in their support. This type of Nash equilibria is called weakly-stable. We note that pure Nash equilibria are trivially weakly-stable. It has been shown that mixed Nash equilibria are not weakly-stable in generic games [2]

We can show that Nash equilibria that are not weakly-stable Nash are actually unstable for GDA. Moreover, through standard dynamical systems machinery, that the set of initial conditions that converges to Nash equilibria that are not weakly-stable should be of Lebesgue measure zero. Formally, we prove that:

**Theorem 3.2** (Non weakly-stable Nash are unstable). *Consider a two-team zero-sum game with the utility function of Team B ($\boldsymbol{y}$ vector) being $U(\boldsymbol{x}, \boldsymbol{y})$ and Team A ($\boldsymbol{x}$ vector) being $-U(\boldsymbol{x}, \boldsymbol{y})$. Moreover, assume that $(\boldsymbol{x}^*, \boldsymbol{y}^*)$ is a Nash equilibrium of full support that is not weakly-stable. It follows that the set of initial conditions so that GDA converges to $(\boldsymbol{x}^*, \boldsymbol{y}^*)$ is of measure zero for step size $\eta < \frac{1}{L}$ where $L$ is the Lipschitz constant of $\nabla U$.*

### 3.3 GENERALIZED MATCHING PENNIES (GMP)

Inspired by Theorem 3.2, in this section we construct a family of team zero-sum games so that GDA, OGDA, OMWU, and EG methods fail to converge (if the initialization is a random point in the simplex, the probability of convergence of the aforementioned methods is zero). The intuition is to construct a family of games, each of which has only mixed Nash equilibria (that are not weakly-stable), *i.e.*, the constructed games should lack pure Nash equilibria; using Theorem 3.2, it would immediately imply our claim for GDA. It turns out that OGDA, OMWU, and EG also fail to converge for the same family.

**Definition of GMP.**  Consider a setting with two teams (Team $A$, Team $B$), each of which has $n = 2$ players. Inspired by the standard matching pennies game and the game defined in (Schulman & Vazirani, 2019a), we allow each agent $i$ to have two strategies/actions that is $S = \{H, T\}$ for both teams with $2^4$ possible strategy profiles. In case all the members of a Team choose the same strategy say $H$ or $T$ then the Team "agrees" to play $H$ or $T$ (otherwise the Team "does not agree").

|  | $HH$ | $HT/TH$ | $TT$ |
|---|---|---|---|
| $HH$ | $1, -1$ | $\omega, -\omega$ | $-1, 1$ |
| $HT/TH$ | $-\omega, \omega$ | $0, 0$ | $-\omega, \omega$ |
| $TT$ | $-1, 1$ | $\omega, -\omega$ | $1, -1$ |

Thus, in the case that both teams "agree", the payoff of each team is actually the payoff for the two-player matching pennies. If one team "agrees" and the other does not, the team that "agrees" enjoys the payoff $\omega \in (0, 1)$ and the other team suffers a penalty $\omega$. If both teams fail to "agree", both teams get payoff zero. Let $x_i$ with $i \in \{1, 2\}$ be the probability that agent $i$ of Team $A$ chooses $H$ and $1 - x_i$ the probability that she chooses $T$. We also denote $\boldsymbol{x}$ as the vector of probabilities for Team $A$. Similarly, we denote $y_i$ for $i \in \{1, 2\}$ be the probability that agent $i$ of Team $B$ chooses $H$ and $1 - y_i$ the probability that she chooses $T$ and $\boldsymbol{y}$ the probability vector.

**Properties of GMP.**  An important remark on the properties of our presented game is due. Existing literature tackles settings with (i) (weak-)monotonocity (Mertikopoulos et al., 2019; Diakonikolas et al., 2021), (ii) cocoercivity (Zhu & Marcotte, 1996), (iii) zero-duality gap (Von Neumann, 1928), (iv) unconstrained solution space (Golowich et al., 2020) . Our game is carefully crafted and – although it has a *distinct structure* and is nonconvex-nonconcave only due to *multilinearity*– satisfies none of the latter properties. This makes the (local) convergence of our proposed method even more surprising. (See also Appendix B.6.)

The first fact about the game that we defined is that for $\omega \in (0, 1)$, there is only one Nash equilibrium $(\boldsymbol{x}^*, \boldsymbol{y}^*)$, which is the uniform strategy, *i.e.*, $x_1^* = x_2^* = y_1^* = y_2^* = \frac{1}{2}$ for all agents $i$.

**Lemma 3.1** (GMP has a unique Nash). The Generalized Matching Pennies game exhibits a unique Nash equilibrium which is $(\boldsymbol{x}^*, \boldsymbol{y}^*) = ((\frac{1}{2}, \frac{1}{2}), (\frac{1}{2}, \frac{1}{2}))$.

---

[2] Roughly speaking, generic games where we add small Gaussian noise to perturb slightly every payoff only so that we preclude any payoff ties. In these games, all Nash equilibria in all but a measure-zero set of games exhibit the property that all pure best responses are played with positive probability.

**Remark 1.** *The fact that the game we defined has a unique Nash equilibrium that is in the interior of $[0,1]^4$ is really crucial for our negative convergence results later in the section as we will show that it is not a weakly-stable Nash equilibrium and the negative result about GDA will be a corollary due to Theorem 3.2. We also note that if $\omega = 1$ then there are more Nash equilibria, in particular the $(\mathbf{0},\mathbf{0}),(\mathbf{1},\mathbf{0}),(\mathbf{0},\mathbf{1}),(\mathbf{1},\mathbf{1})$ which are pure.*

The following Theorem is the main (negative) result of this section.

**Theorem 3.3** (GDA, OGDA, EG, and OMWU fail). *Consider GMP game with $\omega \in (0,1)$. Assume that $\eta_{GDA} < \frac{1}{4}$, $\eta_{OGDA} < \min(\omega, \frac{1}{8})$, $\eta_{EG} < \frac{\omega}{2}$, and $\eta_{OMWU} < \min\left(\frac{1}{4}, \frac{\omega}{2}\right)$ (bound on the stepsize for GDA, OGDA, OMWU, and EG methods respectively). It holds that the set of initial conditions so that GDA, OGDA, OMWU, and EG converge (stabilize to any point) is of measure zero.*

**Remark 2.** *Theorem 3.3 formally demonstrates that the behavior of algorithms mentioned in Section 2.1.2 are not a result of "bad parametrization", and in fact, the probability that any of them converges to the NE is equal to the probability that the initialization of the variables coincides with the NE (Lebesgue measure zero).*

**Remark 3** (Average iterate also fails). *One might ask what happens when we consider average iterates instead of the last iterate. It is a well-known fact (Syrgkanis et al., 2015) that the average iterate of no-regret algorithms converges to coarse correlated equilibria (CCE) so we expect that the average iterate stabilizes. Nevertheless, CCE might not be Nash equilibria. Indeed we can construct examples in which the average iterate of GDA, OGDA, OMWU, and EG experimentally fail to stabilize to Nash equilibria. In particular, we consider a slight modification of GMP; players and strategies are the same but the payoff matrix has changed and can be found reads*

|  | $HH$ | $HT/TH$ | $TT$ |
|---|---|---|---|
| $HH$ | $2,-2$ | $\frac{1}{2},-\frac{1}{2}$ | $-2,2$ |
| $HT/TH$ | $-\frac{1}{2},\frac{1}{2}$ | $0,0$ | $-\frac{1}{2},\frac{1}{2}$ |
| $TT$ | $-1,1$ | $\frac{1}{2},-\frac{1}{2}$ | $1,-1$ |

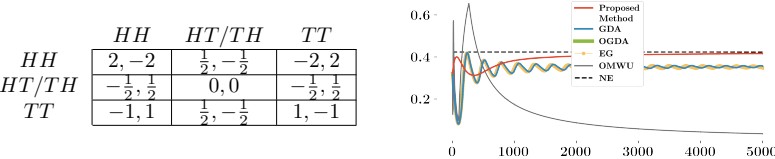

Figure 3: GDA, OGDA, OMWU, & EG fail to converge to a Nash Equilibrium even in average

Figure 3 shows that the average iterates of GDA, OGDA, OMWU, and EG stabilize to points that are not Nash equilibria. We note that since our method (see next subsection) converges locally, the average iterate should converge locally to a Nash equilibrium.

## 3.4 Our proposed method

The aforementioned results prove that the challenging goal of computing two-team zero-sum games calls for an expansion of existing optimization techniques. The mainstay of this effort and our positive result is the KPV-GDA method defined in (KPV-GDA) which is inspired by techniques of adaptive control literature. The first statement we make is that KPV-GDA stabilizes around any Nash equilibrium for appropriate choices of matrices $\mathbf{K}, \mathbf{P}$:

**Theorem 3.4** (KPV-GDA stabilizes). *Consider a team zero-sum game so that the utility of Team B is $U(\boldsymbol{x}, \boldsymbol{y})$ and hence the utility of Team A is $-U(\boldsymbol{x}, \boldsymbol{y})$ and a Nash equilibrium $(\boldsymbol{x}^*, \boldsymbol{y}^*)$ of the game. Moreover, we assume*

$$\left( \begin{array}{cc} -\nabla_{\boldsymbol{x}\boldsymbol{x}}^2 U(\boldsymbol{x}^*, \boldsymbol{y}^*) & -\nabla_{\boldsymbol{x}\boldsymbol{y}}^2 U(\boldsymbol{x}^*, \boldsymbol{y}^*) \\ \nabla_{\boldsymbol{y}\boldsymbol{x}}^2 U(\boldsymbol{x}^*, \boldsymbol{y}^*) & \nabla_{\boldsymbol{y}\boldsymbol{y}}^2 U(\boldsymbol{x}^*, \boldsymbol{y}^*) \end{array} \right) \text{ is invertible.}$$

*For any fixed step size $\eta > 0$, we can always find matrices $K, P$ so that KPV-GDA method defined in (KPV-GDA) converges locally to $(\boldsymbol{x}^*, \boldsymbol{y}^*)$.*

This is an existential theorem and cannot be generally useful in practice. Further, this dynamic would not be necessarily uncoupled and the design of matrices $\mathbf{K}$ and $\mathbf{P}$ could necessitate knowledge of the NE we are trying to compute. Instead, our next statement provides sufficient conditions under which a simple parametrization of matrices $\mathbf{K}, \mathbf{P}$ results in an uncoupled, converging dynamic:

**Theorem 3.5.** *Consider a two-team zero-sum game so that the utility of Team B is $U(\boldsymbol{x}, \boldsymbol{y})$, the utility of Team A is $-U(\boldsymbol{x}, \boldsymbol{y})$, and a Nash equilibrium $(\boldsymbol{x}^*, \boldsymbol{y}^*)$. Moreover, let*

$$\mathbf{H} := \begin{pmatrix} -\nabla^2_{\boldsymbol{xx}}U(\boldsymbol{x}^*, \boldsymbol{y}^*) & -\nabla^2_{\boldsymbol{xy}}U(\boldsymbol{x}^*, \boldsymbol{y}^*) \\ \nabla^2_{\boldsymbol{yx}}U(\boldsymbol{x}^*, \boldsymbol{y}^*) & \nabla^2_{\boldsymbol{yy}}U(\boldsymbol{x}^*, \boldsymbol{y}^*) \end{pmatrix}.$$

*and $E$ be the set of eigenvalues $\rho$ of $\mathbf{H}$ with real part positive, that is $E = \{Eigenvalues\ of\ matrix\ \mathbf{H}, \rho : Re(\rho) > 0\}$. We assume that $\mathbf{H}$ is invertible and moreover*

$$\beta = \min_{\rho \in E} \frac{Re(\rho)^2 + Im(\rho)^2}{Re(\rho)} > \max_{\rho \in E} Re(\rho) = \alpha. \tag{4}$$

*We set $\mathbf{K} = k \cdot \mathbf{I}$, $\mathbf{P} = p \cdot \mathbf{I}$. There exist small enough step size $\eta > 0$ and scalar $p > 0$ and for any $k \in (-\beta, -\alpha)$ so that (KPV-GDA) with chosen $\mathbf{K}, \mathbf{P}$ converges locally to $(\boldsymbol{x}^*, \boldsymbol{y}^*)$.[3]*

## 4 EXPERIMENTS

In this section, we perform a series of experiments to further motivate the study of two-team zero-sum games, especially in the context of multi-agent generative adversarial networks (multi-GANs). A multi-agent generative adversarial network (multi-GAN) (Arora et al., 2017; Hoang et al., 2017; 2018; Zhang et al., 2018; Tang, 2020; Hardy et al., 2019; Albuquerque et al., 2019) is a generative adversarial network (GAN) that leverages multiple "agents" (generators and/or discriminators) in order achieve statistical and computational benefits. In particular, Arora et al. formally proved the expressive superiority of multi-generator adversarial network architectures something that we empirically verify in Section 4. In this direction, researchers strive to harness the efficacy of distributed processing by utilizing shallower networks that can collectively learn more diverse datasets[4].

At first, the superiority of multi-GANs might appear to contrast our theoretical findings; but in reality, the superiority comes from the quality of solutions that are attainable from multi-agent architectures (*expressivity*) and the fact that hardness (*complexity*) translates to rates of convergence but not non-convergence. Single agent GANs quickly converge to critical points that are not guaranteed to capture the distribution very well. In figure 4 we see the fast convergence of a single-agent GAN to solutions of bad quality versus the convergence of a multi-GAN to an obviously better solution. Due to space constraints, we defer further discussion of the experiments at Section D.1.

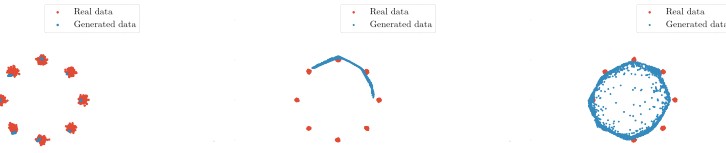

Figure 4: From left to right: (i) Each generator of MGAN learns one mode of the mixture of 8 gaussians, (ii) Mode Collapse of single-agent GANs, (iii) Single-agent GAN can't discriminate between the modes.

## 5 CONCLUSIONS AND OPEN PROBLEMS

In this work we study the wide class of nonconvex-nonconcave games that express the two-team competition, inspired broadly by the structure of the complex competition between multi-agent generators and discriminators in GAN's. Furthermore, in this setting of *two-team zero-sum games*, we have presented a number of negative results about the problem of computing a Nash equilibrium. Moreover, through a simple family of games that we construct, we prove the inability of commonly used methods for min-max optimization such as GDA, OGDA, OMWU, and EG to converge both in average and in the last iterate to Nash Equilibria which comes to a stark contrast to recent literature that is concerned with simpler games. We have also presented an optimization method (called KPV-GDA) that manages to stabilize around Nash equilibria.

---

[3]As long as aforementioned conditions are satisfied, (KPV-GDA) locally converges in any nonconvex-nonconcave game. Indeed, GMP with any $\omega$ satisfies the sufficient conditions of 3.5. See also, Appendix B.7.

[4]Indeed, it is preferable from a computational standpoint to back-propagate through two equally sized neural networks rather than through a single one that would be twice as deep (Tang, 2020).

## ACKNOWLEDGEMENTS

Ioannis Panageas would like to acknowledge a start-up grant. Emmanouil V. VG is grateful for the financial support by FODSI Postdoctoral Fellowship. This work was partially completed while IP and EVVG were visiting research fellows at the Simons Institute for Theory of Computing during Learning and Games Semester.

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
