# OpenReview forum: "Towards convergence to Nash equilibria in two-team zero-sum games"
_ICLR.cc/2023/Conference — ICLR 2023 poster_

### Official Review · Reviewer_m964 · 2022-10-22

**Confidence:** 4
**Correctness:** 3
**Technical Novelty And Significance:** 4
**Empirical Novelty And Significance:** 3
**Recommendation:** 6

**Clarity, Quality, Novelty And Reproducibility:**

The paper is clearly written and addresses a problem previously unsolved in the literature.

**Strength And Weaknesses:**

The premise of the paper is interesting, but I have nontrivial concerns about the technical results. In particular:

1. The CLS-hardness proof seems problematic. The game used in the reduction has $|S_i|$ actions for each of $n$ players, so the normal form of the game has representation size $n \cdot \prod_{i} |S_i|$, which could be exponential in the representation size of the congestion game instance. (Indeed, if we allowed ourselves an algorithm with that runtime, congestion games would become trivial, since one could directly compute the minimizer $\boldsymbol s \in \bigtimes_i S_i$ of the potential!) If the goal is to show that Nash computation is hard for zero-sum team games in some *succinct* game representation, the authors should formally specify exactly what that succinct representation is and why it is interesting. This is my main concern about the paper.

1. I am also unconvinced regarding the solution concept. In particular, in the GAN example (Sec 2.1.1) that seems like a fundamental motivating example in this paper, it seems that the correct solution concept should be TME. The natural goal is to find $G_p$ and $G_\theta$ maximizing the discriminator loss, which is exactly what TME does. Also recall that the gap in solution between an arbitrary Nash and a TME can be arbitrarily bad in general. Especially given that the game is already highly nonconvex-nonconcave, I do not see the benefit of using Nash instead of TME here.

Minor things not affecting my evaluation:

1. The probability expressions after Eq (1) are wrong, since $\zeta$ is a normal variable and therefore $\zeta+p$ may not be in $[0, 1]$.

1. Section 3.2: "[Theorem 3.1] does not preclude the prospect of attaining algorithms (learning first-order methods) that converge to Nash equilibria." I understand that this because you draw the distinction between $\text{poly}(1/\epsilon)$ convergence to $\epsilon$-Nash (which at this point in the paper could still be achieved learning methods) and $\log(1/\epsilon)$ convergence (which would be achieved by an exact method), correct? I think this is worth making explicit, and stating in any hardness result that you are showing hardness for the $\log(1/\epsilon)$ version of convergence.

**Summary Of The Paper:**

The authors investigate the problem of computing Nash equilibrium in (normal-form) two-team zero-sum games. The authors argue that the problem is CLS-hard, and give an algorithm guaranteeing local convergence.

**Summary Of The Review:**

The paper presents new and interesting results on a problem of obvious interest. I have one major concern regarding the technical part of the paper. If it is addressed, I will raise my score.

---

> ### Author Response · Authors · 2022-11-11
> **Re: Official Review of Paper5015 by Reviewer m964 (2/2)**
>
> [1] Goodfellow, I., Pouget-Abadie, J., Mirza, M., Xu, B., Warde-Farley, D., Ozair, S., Courville, A. and Bengio, Y., 2020. Generative adversarial networks. Communications of the ACM, 63(11), pp.139-144.
>
> [2] Borgs, C., Chayes, J., Immorlica, N., Kalai, A.T., Mirrokni, V. and Papadimitriou, C., 2008, May. The myth of the folk theorem. In Proceedings of the fortieth annual ACM symposium on Theory of computing (pp. 365-372).
>
> [3] Daskalakis, C., Fabrikant, A. and Papadimitriou, C.H., 2006, July. The game world is flat: The complexity of Nash equilibria in succinct games. In International Colloquium on Automata, Languages, and Programming (pp. 513-524). Springer, Berlin, Heidelberg.

---

> ### Author Response · Authors · 2022-11-11
> **Re: Official Review of Paper5015 by Reviewer m964 (1/2)**
>
> We thank the reviewer for their very helpful feedback. Below we address your concerns.
>
> >- The CLS-hardness proof seems problematic. The game used in the reduction has actions $|S_i|$ for each of players, so the normal form of the game has representation size $n \cdot \prod_{i} |S_i|$ , which could be exponential in the representation size of the congestion game instance. (Indeed, if we allowed ourselves an algorithm with that runtime, congestion games would become trivial, since one could directly compute the minimizer $\boldsymbol s \in \bigtimes_i S_i$ of the potential!) If the goal is to show that Nash computation is hard for zero-sum team games in some succinct game representation, the authors should formally specify exactly what that succinct representation is and why it is interesting. This is my main concern about the paper.
>
> We thank the reviewer for the insightful comment apologize for the hand-waving writing of our proof. We kindly ask them to check the revised version of our paper in order to assess whether the edited write-up is satisfactory (edits are highlighted).
>
> For our proof, we first assume a *succinctly representable* congestion game [3] that has the *Polynomial Expectation Property*. *I.e.,* the expected cost of each player for any given mixed strategy profile $(s_1, \dots, s_n)$ is computed in time polynomial in:
> * the number agents $n$,
> * $\sum_i|S_i|$ where $S_i$ is the finite set of agent $i$'s strategies,
> * the number of bits required to represent the mixed strategy profile $(s_1, \dots, s_n) $.
>
> These allow for the potential function to be computed in polynomial time and in turn make the constructed two-team zero-sum game succinctly representable and satisfy the polynomial expectation property as well. As [3] note:
> <center><em>"When the number of players is large, the resulting
> computational problems are hardly legitimate, and complexity issues are hopelessly dis-
> torted. This has led the community to consider broad classes of succinctly representable
> games"</em></center>
>
> >- I am also unconvinced regarding the solution concept. In particular, in the GAN example (Sec 2.1.1) that seems like a fundamental motivating example in this paper, it seems that the correct solution concept should be TME. The natural goal is to find $G_p$
> and $G_\theta$ maximizing the discriminator loss, which is exactly what TME does. Also recall that the gap in solution between an arbitrary Nash and a TME can be arbitrarily bad in general. Especially given that the game is already highly nonconvex-nonconcave, I do not see the benefit of using Nash instead of TME here.
>
> The reviewer is correct to point out that the minmax solution (the TME) is the one that we should generally be looking for. In the original GAN paper ([1]), the minmax solution was proven to be the solution of the ''minimax game'' that allows the generator to optimally learn the data distribution. This solution is NP-hard to compute (even in normal-form games [2]) in the general case
> (both in a single-generator GAN and a multiple-generator GAN). Nevertheless, empirical applications of GANs have proven that first-order stationary points of the objective function suffice to produce generated samples of very good quality.
>
> This makes it pretty reasonable for us to relax the solution concept to something more tractable, *namely* a Nash equilibrium.
>
> Notwithstanding, for our illustrative example, every Nash equilibrium of the objective function is also a TME for the generators (since all NE have a unique value). Hence, as we demonstrate in the appendix, it suffices to compute any one of them to learn the parameters of the distribution correctly.
>
> >- The probability expressions after Eq (1) are wrong, since $\zeta$ is a normal variable and therefore may not be in $[0,1]$ .
>
> The reviewer is correct. We should be clearer with our phrasing. Variable $p$ can be interpreted as a probability *only in expectation*. Thank you for pointing it out. Please see the revised version of the paper.
>
> **PS:**
>
> *Further, we kindly prompt the reviewer to take part in our conversation with reviewer 1M7d as they make false claims and they seem to have misunderstood the message and context of our paper which becomes evident when they unjustifiably ask us to compare our work against two particular papers that lie well beyond the scope of our setting, i.e., decentralized learning in repeated games.*

---

> > ### Comment · Reviewer_m964 · 2022-11-17
> > **Response**
> >
> > CLS-hardness: Congestion games are already succinct; that was never the problem. The problem is that the representation of a *two-team zero-sum game* was never specified.
> >
> > Your response does answer my question, though: namely, we should think of normal-form team games as represented by a $\text{poly}(\sum_{i \in \mathcal N} |\mathcal A_i| + \sum_{j \in \mathcal M} |\mathcal B_j|)$-time algorithm that computes the expected utility of a given mixed strategy profile. I would recommend explicitly stating this in the main text, before stating Thm 3.1. I would also recommend including an explicit reference to Daskalakis et al [3], where it is shown that this expected utility problem is easy for congestion games.
> >
> > > Nevertheless, empirical applications of GANs have proven that first-order stationary points of the objective function suffice to produce generated samples of very good quality.
> >
> > This is an important point in demonstrating that the central problem of the paper is interesting. I would recommend stating it explicitly in the paper and including any relevant citation.
> >
> > ---
> >
> > I think Reviewer 1M7d brings up a good point, that I would like to see some more discussion on: namely, regarding the assumption on invertible $\mathbf{H}$ at the equilibrium point. Thm 3.5 assumes that $\mathbf{H}$...
> >
> > 1. is invertible
> > 2. has at least one eigenvalue with positive real part, and
> > 3. satisfies Eq (4). In particular, that implies that either all the eigenvalues with positive real part are equal, or at least one of them has a sufficiently large imaginary part that the equation is satisfied.
> >
> > How reasonable are these assumptions? For example, do all games have at least one equilibrium with this property?
> > Is there some intuitive explanation for what these assumptions are doing? They seem fairly unintuitive, especially the latter two. Why should we expect this condition to hold for a game we may be interested in, for example, the GAN game? The fact that it holds for GMP, which you showed in your response to 1M7d, is comforting at least--I advise including that calculation in the final version so that readers can ground themselves. Certainly, at the very least, the claim "KPV-GDA stabilizes around any Nash equilibrium" (page 8, just before Thm 3.4) is too strong and should be weakened.
> >
> > Similarly, it seems that the parameter $k$ in Thm 3.5, which is required for convergence, requires knowledge of the Nash equilibrium already, since the convergence depends on it being *between* two values $-\beta$ and $-\alpha$, which in turn depend on $\mathbf{H}$. This seems rather circular--more so than, say, the other parameters $\eta$ and $p$ in the same theorem statement, which allow convergence as long as they are "small enough". I would also like some more discussion on this point.

---

> > > ### Author Response · Authors · 2022-11-19
> > > **Re: Response**
> > >
> > > Please check our revised paper to find the edits you suggest:
> > >
> > > >* I would recommend explicitly stating this in the main text, before stating Thm 3.1. I would also recommend including an explicit reference to Daskalakis et al [3], where it is shown that this expected utility problem is easy for congestion games.
> > >
> > > >* This is an important point in demonstrating that the central problem of the paper is interesting. I would recommend stating it explicitly in the paper and including any relevant citation.
> > >
> > > We edited our last draft to include these.
> > >
> > > -----
> > > >* I think Reviewer 1M7d brings up a good point, that I would like to see some more discussion on: namely, regarding the assumption on invertible at the equilibrium point. Thm 3.5 assumes that ...
> > >
> > > The invertibility of matrix $\mathbf{H}$ is pretty standard in an array of works that, like ours, are concerned with local convergence guarantees in min-max optimization.
> > >
> > > Further, the condition on the eigenvalues is *sufficient* and not *necessary*. First of all, it relaxes the conditions for local convergence placed upon GDA, OGDA, EG -- which is evident with our benchmark game, GMP.
> > >
> > > Moreover, in the appendix of the revised paper, we included a number of experiments on randomly generated 2-vs-2 team zero-sum games plotting the Nash equilibrium gap. In virtually all games, the NE-gap converges to $0$ without tuning $\eta, k, p$. This is an indicator that looking into other sufficient conditions of convergence of KPV is an interesting open question for future work!
> > >
> > > We should note that a systematic method of designing washout filters (an instance of which is KPV) is still an open problem for the control theory community.
> > >
> > > >* Is there some intuitive explanation for what these assumptions are doing? They seem fairly unintuitive, especially the latter two.
> > >
> > > From a dynamical systems perspective, the method manages to exploit the oscillatory motion that is generated by the imaginary part of the eigenvalues; this way the trajectories can approach the previously unstable manifold of the dynamic of vanilla GDA.

---

> > > > ### Comment · Reviewer_m964 · 2022-11-19
> > > > **Response**
> > > >
> > > > Okay, that's reasonably comforting. As most of my concerns have been resolved, I will raise my score to 6.
> > > >
> > > > I would highly recommend the authors to include some background, perhaps in the appendix with reasonable pointers throughout the paper, regarding relevant things in dynamical systems for those in your audience (such as me) who do not come from that background--such as the points you made in your response to me and to the other reviewer.

---

> ### Author Response · Authors · 2022-11-17
> **Reminder: Discussion period deadline approaches**
>
> Dear reviewer m964,
>
> Since the deadline of the discussion period approaches --- which means we will no longer be able to revise the manuscript as well --- could you let us know whether the edits we made are satisfactory or not?
>
> Thanks in advance,
>
> The authors.

---

### Official Review · Reviewer_1M7d · 2022-10-25

**Confidence:** 4
**Clarity, Quality, Novelty And Reproducibility:** The above.
**Correctness:** 3
**Technical Novelty And Significance:** 2
**Empirical Novelty And Significance:** 2
**Recommendation:** 3

**Strength And Weaknesses:**

This paper shows that finding a Nash equilibrium in these two-team zero-sum games is CLS-hard and theoretically proposes a new version of gradient-based algorithms, which converges locally to Nash equilibria with some assumptions. However, many things about this algorithm are unclear: the performance relative to existing algorithms, the reason (intuition) why it converges, whether the assumption is realistic in the real world, and how it connects to GAN.
Two-team zero-sum games (Schulman & Vazirani 2019b) have been studied. This paper aims to compute a Nash in these games. Schulman & Vazirani (2019b) proposed using a multilinear program to compute a maxmin equilibrium, which is a Nash equilibrium as well. In addition, there are other algorithms for computing a Nash equilibrium in general multiplayer games [1,2], which can solve two-team zero-sum games. Particularly, the proposed algorithm in this paper is not the first gradient-based algorithm guaranteeing to converge for Nash equilibrium in two-team zero-sum games because gradient-based algorithms with theoretical guarantee in general games were proposed [2], which can solve two-team zero-sum games. It is well-known that a gradient-based algorithm cannot guarantee convergence for Nash equilibrium. For convergence, some assumptions/operations should be added to the algorithm. Then, this paper needs to show that the proposed algorithm is better than existing algorithms in terms of theoretical results and/or experimental results.



Section 3 studies the reason why some settings of gradient-based algorithms cannot converge in GMP. Unfortunately, they only show that their proposed algorithm KPV-GDA can converge in GMP but do not show the reason why their algorithm KPV-GDA can converge in GMP. That is, it is unclear why KP-GDA converges in GMP. Understanding the reason why the proposed algorithm KPV-GDA converges in MPG is also important.


Theorems 3.4 and 3.5 are the main results of this paper about the convergence of the proposed algorithm. However, both theorems assume that the matrix of Jacobian related to the Nash equilibrium is invertible. It is unclear if this assumption is realistic. Particularly, it is not easy for us to know the property of a Nash equilibrium before we can obtain it. That is, before we obtain a Nash equilibrium by using the proposed algorithm, we usually do not know its property. Then this requirement seems unrealistic.

In section 2.1 and Section 4, this paper uses GAN to motivate the study of two-team zero-sum games. However, it is unclear how to model that GAN by using the proposed two-team game. For example, it is unclear what a pure strategy in a Team-WGAN is, and it is unclear what the utility function in a Team-WGAN. I believe continuous games of GAN and discrete games of two-team games have different properties.





[1] Berg, K. and Sandholm, T., 2017, February. Exclusion method for finding Nash equilibrium in multiplayer games. In AAAI.

[2] Gemp, I., Savani, R., Lanctot, M., Bachrach, Y., Anthony, T., Everett, R., Tacchetti, A., Eccles, T. and Kramár, J., 2022. Sample-based Approximation of Nash in Large Many-Player Games via Gradient Descent.  In AAMAS

Minor:

‘lFurthermore’ -> Furthermore

‘a NE’ -> an NE


**Summary Of The Paper:**

This paper studies two-team zero-sum multiplayer games, where there are two teams of players, and team members in each team share the same payoff function. This paper focuses on computing a Nash equilibrium in these games, and they show that finding a Nash equilibrium in these two-team zero-sum games is CLS-hard based on the result of computing a Nash equilibrium in two-team zero-sum games. The result then will follow since computing Nash equilibria in congestion games is CLS-hard (Babichenko & Rubinstein 2021). They show that, with some specific settings, some gradient-based algorithms fail to converge (even locally) to a Nash equilibrium. They then provide a new version of gradient-based algorithms, which converges locally to Nash equilibria with some assumptions.

**Summary Of The Review:**

This paper shows that finding a Nash equilibrium in these two-team zero-sum games is CLS-hard and theoretically proposes a new version of gradient-based algorithms, which converges locally to Nash equilibria with some assumptions. However, many things about this algorithm are unclear: the performance relative to existing algorithms, the reason (intuition) why it converges, whether the assumption is realistic in the real world, and how it connects to GAN.

---

> ### Author Response · Authors · 2022-11-11
> **Re: Official Review of Paper5015 by Reviewer 1M7d (2/2)**
>
> **The following claim is also false**:
> >- Section 3 studies the reason why some settings of gradient-based algorithms cannot converge in GMP. Unfortunately, they only show that their proposed algorithm KPV-GDA can converge in GMP but do not show the reason why their algorithm KPV-GDA can converge in GMP. That is, it is unclear why KP-GDA converges in GMP. Understanding the reason why the proposed algorithm KPV-GDA converges in MPG is also important.
>
> The reason that KPV-GDA converges in GMP is precisely because GMP satisfies the assumptions of Theorem 3.5. If we take matrix $H$, for any $\omega$, the assumptions of Theorem 3.5 are satisfied. It is pretty easy to check it.
>
> For GMP, the eigenvalues of $H$ are:
> $$ \rho \in \{ -2\omega, 2\omega - 2i, 2\omega + 2i \}.$$
>
> The eigenvalues with a positive real part are the following:
> $E = \{ 2\omega - 2i, 2\omega + 2i \}$.
>
> Hence,
> $\beta = \min_{\rho \in E} \frac{\mathrm{Re}(\rho) + \mathrm{Im} (\rho) }{\mathrm{Re}(\rho)} = \frac{2\omega^2 + 2}{\omega} = 2 \omega + \frac{2}{\omega} > 2 \omega = \max_{\rho \in E} \mathrm{Re}(\rho) = \alpha .$
>
> For any choice of $k \in (  -2 \omega - \frac{2}{\omega}, -2\omega)$ the algorithm converges.
>
> For the case that $\omega = 1/2$, it suffices that $k \in (-5, -1)$.
> #### **EDIT:** Check the inclusion of the latter in the appendix.
>
> The **invertibility of the Jacobian matrix is a pretty standard assumption** in theoretical papers:
> >* Theorems 3.4 and 3.5 are the main results of this paper about the convergence of the proposed algorithm. However, both theorems assume that the matrix of Jacobian related to the Nash equilibrium is invertible. It is unclear if this assumption is realistic. Particularly, it is not easy for us to know the property of a Nash equilibrium before we can obtain it. That is, before we obtain a Nash equilibrium by using the proposed algorithm, we usually do not know its property. Then this requirement seems unrealistic.
>
> If the Jacobian matrix is not invertible gradient descent is guaranteed to not converge either! The assumption of invertibility is rather standard in a number of works in minmax optimization and/or learning in games ([4; Assumption 1], [5; Assumption 1.7], [6; Statement of Theorem 3]).
>
> The values of scalars $k,p$ are subject to hyper-parameter optimization which is pretty standard in the machine learning community. This is the case even for the learning rate of gradient descent when used for the purpose to train a neural net.
>
> Further, the **assumptions of Theorem 3.5 are only sufficient and not necessary**.
>
> #### **EDIT:** We further included experiments in randomly generated two-team zero-sum games in Section E of our appendix in which our proposed method converges to NE without any parameter tuning taking place for each one of these games. This showcases the fact that the conditions we posit in Thrm 3.5 are indeed *sufficient* and not necessary. Nevertheless, **they are less restrictive** than conditions needed by GDA, OGDA, EG, OMWU to converge!
>
> >* In section 2.1 and Section 4, this paper uses GAN to motivate the study of two-team zero-sum games. However, it is unclear how to model that GAN by using the proposed two-team game. For example, it is unclear what a pure strategy in a Team-WGAN is, and it is unclear what the utility function in a Team-WGAN. I believe continuous games of GAN and discrete games of two-team games have different properties.
>
> It is true that Team-WGAN resembles a continuous game rather than a discrete one and that discrete games are different than continuous in their properties. Nevertheless, the algorithms used to optimize GANs are effectively variations of gradient descent-ascent. Our proposed method is guaranteed to locally converge under the assumptions we state. Further, it is theoretically guaranteed to converge even cases that gradient descent-ascent does not. Further, our method is empirically verified to converge when GDA does not in our paper. And most certainly, the algorithms of papers ([1], [2]) the reviewer mentions are of no use for this setting.
>
>
>
>
> [1] Berg, K. and Sandholm, T., 2017, February. Exclusion method for finding Nash equilibrium in multiplayer games. In AAAI.
>
> [2] Gemp, I., Savani, R., Lanctot, M., Bachrach, Y., Anthony, T., Everett, R., Tacchetti, A., Eccles, T. and Kramár, J., 2022. Sample-based Approximation of Nash in Large Many-Player Games via Gradient Descent. In AAMAS
>
>
> [3] Cesa-Bianchi, N. and Lugosi, G., 2006. Prediction, learning, and games. Cambridge university press.
>
> [4] Wang, Y., Zhang, G. and Ba, J., 2019. On solving minimax optimization locally: A follow-the-ridge approach. ICLR 2020.
>
> [5] Daskalakis, C. and Panageas, I., 2018. The limit points of (optimistic) gradient descent in min-max optimization. Advances in neural information processing systems, 31.
>
> [6] Fiez, T. and Ratliff, L., 2020. Gradient descent-ascent provably converges to strict local minmax equilibria with a finite timescale separation. ICLR 2021.

---

> ### Author Response · Authors · 2022-11-11
> **Re: Official Review of Paper5015 by Reviewer 1M7d (1/2)**
>
> We are happy to include the papers mentioned by the reviewer (please check the revised version of our paper).  Nevertheless, we note that **neither of those papers concerns online, decentralized algorithms**. We are very clear w.r.t. the scope of our study; *i.e.,* **we study no-regret and online algorithms for learning in games** and not any algorithm for solving games within the ([3; Chapter 7]) *repeated game* framework. We do not really have to argue about the merit of *online learning* and the *repeated game* framework which are standard and well-established.
>
> Comparing no-regret online algorithms for learning in games with the heuristic presented in the paper mentioned by the reviewer ([2]) ---
> just because there is some sort of first-order gradient information involved in both of them--- makes little to no sense.
>
>
> >- However, many things about this algorithm are unclear: the performance relative to existing algorithms, the reason (intuition) why it converges, whether the assumption is realistic in the real world, and how it connects to GAN.
>
> The **reason** that KPV-GDA converges is the one demonstrated in Theorem 3.5. In the case that every player follows a certain online learning algorithm (GDA, OGDA, EG, KPV-GDA) it gives rise to a dynamical system. This dynamical system can have fixed points that are either unstable, stable, or asymptotically stable. All of the fixed points of this dynamical system are Nash equilibria of the game (this is folklore). KPV-GDA manages to make certain fixed points asymptotically stable even when they were unstable for the dynamical systems induced by GDA, OGDA, or EG. This translates to local convergence of KPV-GDA to a Nash equilibrium.
>
> Further, we do make a **comparison** to existing algorithms throughout the paper. Of course, we compare our method against algorithms that work within the framework of a *repeated game*, or *online learning* with *full-information feedback*. *I.e.,* a finite number of players with a finite number of actions repeatedly plays the same game for multiple rounds. At every round, each player commits to a certain strategy and observes the expected loss for their given strategy as well as the gradient of their w.r.t. their strategy vector. It is obvious that the algorithms of references [1], [2] do not work within this framework.
>
>
> Further, we cannot help but notice some **false claims in this review**, *e.g.*:
>
>
> >- Two-team zero-sum games (Schulman & Vazirani 2019b) have been studied. This paper aims to compute a Nash in these games. Schulman & Vazirani (2019b) proposed using a multilinear program to compute a maxmin equilibrium, which is a Nash equilibrium as well.
>
> **(Schulman & Vazirani 2019b) propose no algorithm for the computation of any equilibrium**, let alone a team maxmin equilibrium.
>
>
> Also, nowhere in the paper do we suggest that we propose the first algorithm for solving two-team zero-sum games:
>
> >- In addition, there are other algorithms for computing a Nash equilibrium in general multiplayer games [1,2], which can solve two-team zero-sum games. Particularly, the proposed algorithm in this paper is not the first gradient-based algorithm guaranteeing to converge for Nash equilibrium in two-team zero-sum games because gradient-based algorithms with theoretical guarantee in general games were proposed [2], which can solve two-team zero-sum games. It is well known that a gradient-based algorithm cannot guarantee convergence for Nash equilibrium. For convergence, some assumptions/operations should be added to the algorithm. Then, this paper needs to show that the proposed algorithm is better than existing algorithms in terms of theoretical results and/or experimental results.
>
> We never claimed we propose the first algorithm ever to compute a Nash equilibrium in two-team zero-sum games. There exists an array of centralized approximation algorithms for multi-player games. It is obvious though that those algorithms do not fit within the framework of *online-learning with full-information feedback* (please refer to the cornerstone reference [3; Chapter 7]).
>  Throughout the paper, we are very careful to be clear that we are interested in examining the performance of online learning algorithms. The fact that the algorithm of [2] uses the first-order derivative of *some* function does not mean it is an online, learning algorithm.

---

### Official Review · Reviewer_fCR8 · 2022-10-26

**Confidence:** 2
**Correctness:** 4
**Technical Novelty And Significance:** 3
**Empirical Novelty And Significance:** Not applicable
**Recommendation:** 6

**Clarity, Quality, Novelty And Reproducibility:**

The introduction is well written, covering an extensive literature on game theory (different solution concepts and how they compare, importance of Nash Equilibrium in applications such as security, networks, and evolution). The technical exposition is also well written, if somewhat a bit too brief.

All theoretical results are new, and some of them require simple, but slightly new perspectives.

The simple experiments are described in enough details to reproduce.

**Strength And Weaknesses:**

What I like:
* The reduction to show CLS-hardness is very simple.
* The counter example (Generalized Matching Pennies) showing non-convergence is very simple and intuitive.
* The motivation for the new first-order method (KPV_GDA) to stabilize Nash Equilibria in certain general class of well-behaved games is also intuitive.

What might be improved:
* The argument connecting (the results in this paper concerning) two-team zero-sum games with (the real world performance of) multi-agent GANs, while interesting, may need stronger evidence.

**Summary Of The Paper:**

This paper studies the computation of Nash Equilibrium for two-team zero-sum games, and

1. proves that computing an approximate (possibly mixed) NE in two-team zero-sum games is CLS-hard.

2. gives a simple family of two-team zero-sum games where some common online, first-order min-max optimization methods (GDA, OGDA, OMWU, EG) fail to converge to NE (regardless of convergence time) except for a measure zero set of initial conditions.

3. gives a new first-order method (KPV-GDA) to stabilize the dynamics around unstable Nash Equilibria in certain class of well-behaved games and converge to them.

4. argues that the study of two-team zero-sum games applies to multi-agent GANs, which performs better than a single-agent GAN on learning a mixture of 8 Gaussians in an experiment.

**Summary Of The Review:**

The paper contributes to the study of two-team zero-sum games, and this reader recommends acceptance.

---

> ### Author Response · Authors · 2022-11-11
> **Re: Official Review of Paper5015 by Reviewer fCR8**
>
> We thank the reviewer for their time and comments. Please, let us know whether we could clarify anything else.
>
> *Further, we kindly prompt the reviewer to take part in our conversation with reviewer 1M7d as they make false claims and they seem to have misunderstood the message and context of our paper which becomes quite evident when they unjustifiably ask us to compare our work against two particular papers that lie well beyond the scope of our setting, i.e., decentralized learning in repeated games.*

---

### Author Response · Authors · 2022-11-19
**Last revisions**

Dear reviewers,

We have made the last edits to our draft. We fixed some typos and mainly:

* included a remarking comment as to why a NE equilibrium is a reasonable solution concept for the team-WGAN;
* added an extra section in the appendix where we apply our method without parameter tuning on randomly generated two-team zero-sum games. This demonstrates that the conditions on Thm 3.5 are only sufficient and investigating less restrictive sufficient conditions for convergence is a fascinating future direction;
* we made clear that two-team zero-sum games are assumed to be succinctly representable and change theorem 3.1 accordingly.

Let us know if you have further feedback. We hope we managed to address your concerns.

---

### Decision · Program_Chairs · 2023-01-20

**Decision:**

Accept: poster

**Justification For Why Not Higher Score:**

I have the sense from all of the reviewers that the motivation behind this work needs to be clearer.

**Justification For Why Not Lower Score:**

The technical content seems solid, especially after a productive exchange with one reviewer.  The negative reviewer's concerns primarily revolve around the need to compare to batch benchmarks, but I agree with reviewer m964 that this is not necessary in a paper that aims to demonstrate feasibility for an online algorithm.



**Metareview: Summary, Strengths And Weaknesses:**

(a) Summary: This paper proves that finding Nash equilibrium in two-team zero-sum games is CLS-hard in general, exhibits a class of simple games for which common online algorithms fail to converge, and proposes an algorithm that does converge to a Nash equilibrium under certain assumptions.  A connection to training GANs is proposed.

(b) Strengths:  One reviewer found the three main results (CLS-hardness proof, demonstration of non-convergence, and demonstration of a well-behaved class of games) clearly expressed.  The other positive reviewer was eventually satisfied on these points after an detailed conversation with the authors.

(c) Weaknesses:  All reviewers found the motivating connection to training GANs rather unconvincing.  One reviewer felt strongly that non-online benchmarks ought to have been included.

**Note From Pc:**

if the above contains the word "oral" or "spotlight" please see: "oral" presentation means -> notable-top-5% and "spotlight" means -> notable-top-25%. As stated in our emails, we are disassociating presentation type from AC recommendations

**Summary Of Ac-Reviewer Meeting:**

n/a